# Evaluation of Microvascular Density in Glioblastomas in Relation to p53 and Ki67 Immunoexpression

**DOI:** 10.3390/ijms25126810

**Published:** 2024-06-20

**Authors:** Tamás-Csaba Sipos, Attila Kövecsi, Lóránd Kocsis, Monica Nagy-Bota, Zsuzsánna Pap

**Affiliations:** 1Department of Anatomy and Embryology, “George Emil Palade” University of Medicine, Pharmacy, Sciences and Technology of Târgu Mures, 540142 Târgu Mures, Romania; tamas.sipos@umfst.ro (T.-C.S.);; 2Doctoral School of Medicine and Pharmacy, “George Emil Palade” University of Medicine, Pharmacy, Sciences and Technology of Targu Mures, 540142 Targu Mures, Romania; 3Pathology Department, County Emergency Clinical Hospital of Târgu Mureș, 540136 Târgu Mureș, Romania; 4Pathology Department, “George Emil Palade” University of Medicine, Pharmacy, Sciences and Technology of Târgu Mures, 540142 Târgu Mures, Romania

**Keywords:** CD34, CD105, angiogenesis, *IDH1*, glioblastoma, p53, Ki67

## Abstract

Glioblastoma is the most aggressive tumor in the central nervous system, with a survival rate of less than 15 months despite multimodal therapy. Tumor recurrence frequently occurs after removal. Tumoral angiogenesis, the formation of neovessels, has a positive impact on tumor progression and invasion, although there are controversial results in the specialized literature regarding its impact on survival. This study aims to correlate the immunoexpression of angiogenesis markers (CD34, CD105) with the proliferation index Ki67 and *p53* in primary and secondary glioblastomas. This retrospective study included 54 patients diagnosed with glioblastoma at the Pathology Department of County Emergency Clinical Hospital Târgu Mureș. Microvascular density was determined using CD34 and CD105 antibodies, and the results were correlated with the immunoexpression of *p53*, *IDH1*, *ATRX* and Ki67. The number of neoformed blood vessels varied among cases, characterized by different shapes and calibers, with endothelial cells showing modified morphology and moderate to marked pleomorphism. Neovessels with a glomeruloid aspect, associated with intense positivity for CD34 or CD105 in endothelial cells, were observed, characteristic of glioblastomas. Mean microvascular density values were higher for the CD34 marker in all cases, though there were no statistically significant differences compared to CD105. Mutant *IDH1* and *ATRX* glioblastomas, wild-type *p53* glioblastomas, and those with a Ki67 index above 20% showed a more abundant microvascular density, with statistical correlations not reaching significance. This study highlighted a variety of percentage intervals of microvascular density in primary and secondary glioblastomas using immunohistochemical markers CD34 and CD105, respectively, with no statistically significant correlation between evaluated microvascular density and p53 or Ki67.

## 1. Introduction

Diffuse gliomas, originating from glial stem or progenitor cells [1,2], represent the most frequent tumors of the central nervous system in adults [3]. Glioblastomas, known as astrocytic glioma, grade 4, according to the WHO CNS classification [4], represent the most aggressive and lethal form of primary intracranial malignant tumor in adults [5]. The incidence rate of glioblastomas is 3.19–4.17/100,000 person-years [6], accounting for 50.1% of all malignant tumors of the central nervous system in the USA [7]. The median survival rate ranges between 12 and 15 months after diagnosis [8,9], with the median age of patients being approximately 64 years [10], and a higher incidence in men (men/women = 1.5:1) [11].

The diagnosis of glioblastomas, according to WHO CNS, is based on histological and molecular criteria, classified into the category of diffuse astrocytomas with the highest degree of malignancy (grade 4). Tumor grading is subject to histopathological characteristics such as increased cell density, cyto-nuclear atypia, increased mitotic activity, microvascular proliferation, and necrosis. The presence of at least one of the latter two criteria is mandatory to define grade 4 [12].

Depending on the presence of isocitrate dehydrogenase mutation 1 (*IDH1*), according to WHO 2016, glioblastomas can be subclassified into primary glioblastoma or the wild type (*IDH1* wild type), representing approximately 90% of cases, predominantly in elderly patients, and secondary glioblastoma or *IDH1* mutant (less than 10% of cases), developed on the background of a low-grade glioma, which preferably occurs in young individuals [13], associated with a more favorable survival rate [14].

Glioblastomas have rich but inefficient vascularization, characterized by hypoxia. Hypoxia and inadequate nutrient supply favor the appearance of angiogenic factors, such as vascular endothelial growth factor (VEGF) or platelet-derived growth factor (PDGF), leading to the formation of new vascular networks [15]. Tumoral angiogenesis represents the development of new blood vessels and has been recognized as a distinctive sign of malignant tumors [16]. The degree of angiogenesis, studied as microvascular density (MVD), impacts the progression and invasive nature of the tumor [17,18]. It is now recognized that tumors present alternative mechanisms of vascularization, such as vascular mimicry and the transdifferentiation pathway of tumor cells into endothelial cells. Vascular mimicry represents a model of functional microcirculation generated by tumor cells, lacking endothelial lining, demonstrated by immunohistochemical studies with various endothelial markers, such as CD34 or CD105 [19,20,21]. Anti-angiogenic treatments (bevacizumab) and surgical resection followed by temozolomide and radiotherapy have not achieved the expected effect and have not contributed to improving patient survival in the case of glioblastomas [21,22].

CD34 is a marker of endothelial progenitor cells, which plays a crucial role in regulating angiogenesis in glioblastomas. CD34 stimulates the development of a new network of blood vessels and promotes tumor proliferation and invasion, thus playing a role in worsening the prognosis [23]. Increased CD34 expression in diffuse gliomas has been associated with tumor grade, with the CD34 expression in glioblastomas being higher than in low-grade gliomas (LGGs). However, no correlations have been described between CD34 and patient age or sex [23].

CD105 or endoglin is a transmembrane protein located on the membrane of endothelial cells involved in the angiogenesis of immature vessels. Endothelial cells with active proliferative capacity show increased CD105 expression, which plays an essential role in controlling angiogenesis in glioblastomas, stimulating the development of new vascular networks [24,25]. According to the literature data, the role of increased expressions of biomarkers involved in glioblastoma angiogenesis, such as CD34 or CD105, is currently not fully elucidated [23,24].

Regarding microvascular density in a potential relationship with the Ki67 proliferation index and *p53* mutation, data are controversial [17]. Recent studies have shown that up to 60% of tumor endothelial cells express *p53* protein concomitantly with glial tumor cells in glioblastoma. These results, together with other somatic mutations in the primary tumor, support the idea of transdifferentiation of endothelial cells from adjacent tumor glial cells [18].

The aim of this study was to examine angiogenesis through tumor microcirculation. The microcirculation of glioblastomas was evaluated by histological quantification of the pan-endothelial marker expression, CD34, as well as the newly formed microvessel marker, CD105—endoglin. The immunohistochemical results obtained were correlated with the Ki-67 proliferation index and *p53* protein immunoexpression, respectively, with *IDH1* and alpha-thalassemia/intellectual disability, X-linked (*ATRX)* mutational status.

## 2. Results

We included in our retrospective study 54 cases of glioblastomas. Regarding gender distribution, we found a slight predominance in males (29/54), with a sex ratio of 1.16, favoring males. The majority of patients were over 50 years old (35/54, 64.8%). Regarding laterality, both cerebral hemispheres were equally affected. Most cases were found to be IDH1 wild-type glioblastomas (47/54, 87%), ATRX wild type (39/54, 72%), and p53 wild type (43/54, 79%), with a Ki67 index value over 20% present in 35.19% (19/54) of cases.

MVD-CD34 and MVD-CD105

Regarding tumor vascularization in glioblastomas, we observed that the number of neoformed blood vessels varied from case to case; the vessels were characterized by different shapes and calibers; endothelial cells showed modified morphology, associated with moderate to marked pleomorphism. Characteristic of glioblastomas, we also observed neovessels with a glomeruloid aspect associated with intense positivity for CD34 or CD105 in endothelial cells (Figure 1 and Figure 2).

In some cases, the immunoexpression of CD34 or CD105 in tumor endothelial cells was absent, and these markers did not highlight the analyzed vascular structures. Likely, these vascular structures support the existence of the vascular mimicry mechanism. Instead, some glial cells, likely tumor progenitor cells or those undergoing endothelial transdifferentiation, show positive immunostaining for CD34 (Figure 3). These tumor cells are located around neovessels, suggesting an active proliferation of vascular structures in the adjacent tumor stroma (Figure 4).

The values of microvascular density quantified by the percentage of endothelial cells marked with CD34 ranged from 0.35% to 16.89%, representing all vessels in the tumor stroma, with an average value of 4.13%. The values of microvascular density determined by the CD105 antigen were within similar ranges as CD34, ranging from 0.33% to 19.32%, but with an average value of 3.76%. In our observations, the median vascular density in normal brain tissue was 0.09% as determined by CD34 immunohistochemical staining, and 0.04% as evaluated by CD105 (Figure 5 and Figure 6).

We could not demonstrate statistically significant differences between MVD-CD34 and MVD-CD105 values (*p* = 0.58), but compared to a normal brain, microvascular density is significantly higher in glioblastomas (*p* < 0.0001) (Figure 6). Only in 22.22% (12/54) of cases were higher levels than 5% of positive CD105 vascular areas recorded from the total examined tumor, and in the case of the CD34 marker, only in 31.48% (17/54) of cases.

Regarding MVD-CD34 in the right cerebral hemisphere, most cases had values below 2% (12/27), while in tumors located in the left hemisphere, values above 5% predominated (11/27). MVD-CD34 values above 5% were more frequent in the temporal lobe (36.33%, 8/22) and frontal lobe (50%, 8/16), while those below 2% were more common in the parietal lobe (6/9) and temporal lobe (7/22). However, the highest MVD-CD34 values were recorded in the temporal lobe and the right hemisphere. In the parietal lobe (9/9) and occipital lobe (6/7), MVD-CD34 values were predominantly below 5%. It can be observed that involvement of the temporal and frontal lobes is associated with higher microvascular density compared to the parietal and occipital lobes, but we could not demonstrate a statistically significant association regarding MVD-CD34 and tumor location or laterality (*p* = 0.17, *p* = 0.34) (Table 1).

In both sexes, in most cases, MVD-CD34 values were below 5%. The male-to-female ratio in cases with MVD-CD34 values below 5% was 1.17 (20/17), while in those with MVD-CD34 values above 5%, it was 1.12 (9/8). Most patients over 65 years old (55.5%, 10/18) recorded MVD-CD34 values below 2%, while most lesions with values above 5% developed in patients under 50 years old (31.8%, 7/19). Also, we could not demonstrate a statistically significant association between MVD-CD34 and the age and sex of the patients (*p* = 0.24 and *p* = 0.97, respectively) (Table 1).

In the case of endoglin (CD105), we observed that MVD values ranging from 2% to 5% were more frequent in the left hemisphere (12/27), while in the right hemisphere, cases with MVD-CD105 values below 2% predominated (12/27). In the left hemisphere, cases with MVD-CD105 values above 5% were twice as frequent as in the contralateral hemisphere, but the highest value of MVD-CD105 was determined in a glioblastoma located in the right hemisphere. Regarding location, MVD-CD105 values above 5% were more frequent in the temporal lobe (7/22), while MVD-CD105 values below 2% were more frequent in the parietal lobe (6/9). MVD-CD105 values ranging from 2% to 5% were more frequent in the temporal lobe (11/22). The highest value of MVD-CD105 was recorded in the frontal lobe and the right hemisphere. No statistically significant association was observed between MVD-CD105 and tumor location and laterality (*p* = 0.10, *p* = 0.26) (Table 1, Figure 7).

MVD-CD105 values ranging from 2% to 5% were more frequent in patients over 50 years old (17/35), compared to patients under 50 years old, who more frequently presented MVD-CD105 values below 2% (7/19). MVD-CD105 values below 5% were more frequently recorded in males, with a male-to-female ratio of 1.21 (23/19). No statistically significant association was observed between MVD-CD105 and the sex and age of the patients (*p* = 0.31, *p* = 0.54) (Table 1).

Cases with IDH1 mutations present higher median microvascular density both through the CD34 marker, 3.75% (1.89*–*6.57), and through CD105, 3.92% (0.8*–*5.9), compared to cases where the mutation was not present (2.76% (1.48*–*6.05) and 2.89% (1.38*–*4.63), respectively). Thus, IDH1 mutant glioblastomas exhibit a more abundant and prolific microvascular density compared to wild-type ones, but the difference between these values is not statistically significant (*p* = 0.50, *p* = 0.74). It is worth mentioning that microvascular density, both in primary and secondary glioblastomas, highlighted by CD34 and CD105 markers, recorded nearly equal mean values, with the ratio of MVD-CD105/MVD-CD34 in IDH1 wild-type and mutant tumors being 1.04 (Table 2).

Most cases of the IDH1 wild type present MVD-CD34 values below 2% (19/37), while most IDH1 mutant-type glioblastomas have MVD-CD34 values above 5% (3/7). Regarding CD105, cases with microvascular density ranging from 2% to 5% were found to be the most frequent among IDH1 wild-type glioblastomas (22/47). Cases with the highest vascular microdensities through CD105 were IDH1 wild-type glioblastomas, whereas the highest MVD-CD34 value was observed in the case of an IDH1 mutant glioblastoma. Additionally, the association between the expression of IDH1 markers and CD34 or CD105 did not prove to be statistically significant (*p* = 0.7, *p* = 0.2) (Figure 8, Table 1).

In cases of ATRX mutant type, we observed higher median values of microvascular density both through the CD34 marker, 3.75% (2.11*–*7.69), and through CD105, 3.92% (1.03*–*6.04). In cases where the ATRX mutation was present, MVD-CD105 was slightly higher compared to MVD-CD34. We could not demonstrate statistically significant differences between the mean values of MVD-CD34 and MVD-CD105 in relation to the ATRX mutation (*p* = 0.10 and *p* = 0.39, respectively) (Table 2). The ratio of cases with MVD-CD34 below 2% was higher in ATRX wild-type glioblastomas (17/39), while the ratio of those with MVD-CD34 above 5% was higher in ATRX mutant glioblastomas (6/15). Most ATRX wild-type glioblastomas exhibit MVD-CD105 between 2 and 5% (19/39). There was no statistically significant association observed between ATRX and microvascular density analyzed through CD34 and CD105 markers (*p* = 0.5 and *p* = 0.12, respectively) (Table 1).

Regarding microvascular density in relation to the p53 mutation, the median values of microvascular density were higher in wild-type glioblastomas, with higher values in MVD-CD105. The ratio between the median values of CD105-positive microvascular density among wild-type p53 tumors (2.9% (1.36*–*4.84)) and mutant p53 tumors (2.12% (1.48*–*4.87)) was 1.36, and for MVD-CD34, it was 1.09. Mutant p53 cases showed a more pronounced microvascular density through CD34 expression (2.6% (1.53*–*6.86)) compared to CD105 expression (2.12% (1.48*–*4.87)), while in wild-type p53 glioblastomas, the determined microvascular density was approximately equal through both CD105 (2.9% (1.36*–*4.84)) and CD34 (2.84% (1.36*–*6.05)). There was no statistically significant difference between the mean values of MVD through CD34 or CD105 in both mutant- and wild-type p53 cases (*p* = 0.57 and 0.96, respectively) (Table 2). Regarding MVD-CD34 with values below 2%, it was more frequent in wild-type p53 glioblastomas (17/43). MVD-CD105, with values of 2*–*5%, was more frequent in wild-type p53 tumors (20/43). In contrast, mutant p53 cases with MVD-CD105 values below 2% (5/11) were more frequent compared to MVD-CD34 (4/11) (Table 1). It can be observed that through both MVD-CD34 and MVD-CD105, the highest values were recorded in patients with wild-type p53 glioblastomas (Figure 8).

Regarding the Ki67 index, vascular proliferation through CD34 and CD105 immunoreactivity recorded higher median values in cases where the Ki67 index was above 20% compared to those below 5%. In cases with Ki67 below 5%, microvascular density showed higher proliferation through CD34 immunoreactivity (2.56 (1.16*–*3.89) compared to CD105 (2.39 (0.76*–*5.15). The highest median values of microvascular density, through CD34, 3.62% (1.57*–*8.63), were recorded in cases where the Ki67 index ranged between 5 and 20%, in contrast to CD105 immunoreactivity, where the median microvascular density was 2.39% (1.95*–*3.99) (Table 2).

Cases where the Ki67 index was over 20% showed a more pronounced microvascular density through CD105 (2.99 (1.73*–*4.84)) compared to CD34 (2.76 (1.48*–*6.05)); however, the difference between the mean values of MVD-CD34 and MVD-CD105 within the studied Ki67 proliferation index intervals was not statistically significant (*p* = 0.39 and *p* = 0.7, respectively) (Table 2, Figure 9).

Among glioblastomas with MVD-CD34 under 5%, those with a Ki67 proliferative index under 5% were more frequent (15/37). In cases with MVD-CD34 over 5%, in the majority of cases (14/17), a Ki67 proliferative index over 5% was observed. A Ki67 proliferative index over 5% was observed in the majority of cases with MVD-CD105 under 2% (10/19), in 82.6% (19/23) of cases with MVD-CD105 between 2 and 5%, and in 58.3% (7/12) of cases with MVD-CD105 over 5%. We could not demonstrate a statistically significant association between the Ki67 index and the microvascular density markers CD34 and CD105 (*p* = 0.38 and *p* = 0.26, respectively) (Table 1).

## 3. Discussion

In this retrospective immunohistochemical study, we included 54 cases of glioblastomas, which predominantly developed in patients over 65 years old with a left hemisphere predominance. We correlated MVD-CD34 and MVD-CD105 with clinicopathological parameters and the immunohistochemical results obtained through *p53*, Ki67, *IDH1*, and *ATRX* antibodies. Most cases were found to be the *IDH1* wild type, *ATRX,* and *p53* wild type, with a Ki67 index over 20%. The mean values of microvascular density were higher for the CD34 marker; however, comparing them with those obtained for CD105, there were no statistically significant differences between them. We found that *IDH1* and *ATRX* mutant glioblastomas, wild-type *p53*, and those with a Ki67 index over 20% exhibit a more abundant and prolific microvascular density, with statistical correlations not reaching significant values.

Globally, primary brain tumors represent the 17th most common type of cancer, with approximately 77% of them being of glial origin [26]. Glioblastomas are one of the leading causes of tumor-related mortality worldwide. Molecularly, the mutational status of isocitrate dehydrogenase (*IDH1*) has been demonstrated as a prognostic factor in primary glioblastomas but does not represent a predictive factor for immunotherapeutic treatment [27]. According to data from the literature, the number of *IDH1* mutant cases varies considerably. Similarly to our case series, in studies conducted by Martinez-Lage et al. and Munthe et al., the presence of the mutation ranged between 4.1 and 11.5% of cases, confirming the data predicted by WHO [28,29]. In contrast to our results, in the literature, the number of *IDH1* mutant glioblastomas varies between 22.91 and 38.5% [30,31,32]. In the study conducted by Deacu et al., the patients’ survival was not influenced by the presence or absence of the *IDH1* mutation [33]. Additionally, there is still insufficient data regarding the relationship between the *IDH1* mutation status of the tumor and tumor angiogenesis [34].

The cellular and molecular mechanisms of angiogenesis in glioblastomas are currently strong points in the research field of new therapeutic targets [35]. The mechanism of angiogenesis or vasculogenesis in glioblastomas has been described in numerous studies, but there are still considerable controversies. The cells responsible for tumor neovascularization can be endothelial cells derived from bone marrow or progenitor stem cells, through mechanisms not yet fully elucidated [36]. Microvascular density can be an unfavorable prognostic factor in malignant gliomas [37].

The heterogeneity of vascular morphology in glioblastomas is represented by a variety of vascular patterns relevant to clinical prognosis. In the study by Chen et al., microvascular density was studied based on four types of vascular patterns. The number of CD34-positive cells in microvascular sprouting (MS) and vascular cluster (VC) patterns was significantly lower than that in vascular garland (VG) and glomeruloid vascular patterns (GVPs), with median values of CD34 immunostaining ranging from 5.91 to 10.29 [34]. In the study conducted by Jha et al., the microvascular density measured by CD34 ranged from 9.2% to 41.9% (HPF) and showed a statistically significant association with Ki-67 expression, unlike our results [17]. In the study by Clara et al., the mean value of microvascular density studied by CD34 was 23.9%, while for CD105 it was only 8.9%, with the former being statistically significantly correlated with HIF, unlike our case where microvascular density was on average lower. The CD34/CD105 ratio was 2.68, higher than our results (1.06) [38].

Moghaddam et al. recorded mean values of microvessel density with CD105 in neoplastic areas of 14.28%, compared to non-neoplastic areas, with a significant difference between them (*p* = 0.012). The mean expression of the proliferation index (Ki-67) was 21.44%, with both markers, CD105 and CD31, correlating with Ki67 immunostaining, unlike our results [39]. Similar to our study, Mikkelsen et al. demonstrated that CD105-MVD did not significantly correlate with endothelial cell density, with a mean value of 16.5% [40]. McGahan et al., in their immunohistochemical study evaluating microvascular density, described a positive correlation between CD105, CD34 expression, and tumor-associated hemorrhage [41]. Tamma et al., in their study, showed that *p53*-negative tumor cells are positively correlated with CD34-positive endothelial cells. These data may confirm that the presence of immune and inflammatory cells in the tumor microenvironment contributes to tumor progression and angiogenesis [26]. Tamma et al. analyzed CD34 immunoexpression in normal cerebral tissue compared to glioblastomas to establish differences in microvascular densities. They observed a significant increase in MVD-CD34 in tumor tissue, with a median value of 2.1%, compared to normal brain tissue, which had a median value of 0.58%. In our study, the median value of MVD-CD34 in glioblastoma tissue was 2.8%, while in normal brain tissue, it was 0.09% [26]. The tumor microenvironment comprises several components. In our study, we analyzed the microvascular density through CD34 or CD105 immunostaining, in comparison with the presence or absence of *p53* mutation in glial tumor cells, and found no statistically significant correlation. In the study conducted by Tamma et al., where *p53* positivity was higher in tumor cells compared to normal brain tissue, no correlation was found between the tumor microenvironment and *p53* mutation [26].

In the study conducted by Alkhaibary et al., the Ki-67 index was proposed as a prognostic factor, but previous studies showed contradictory results. Regarding the importance of the Ki-67 proliferation index, it has been shown that a higher Ki-67 index predisposes to longer survival. Alkhaibary et al. found no statistically significant correlation between the Ki-67 index and survival rate [42].

Similar to our study, Bastos et al. demonstrated higher levels of CD105 and Ki-67, which seem to be associated with more aggressive glioblastomas, but they did not record any statistically significant association between the two markers [43]. Burghardt I et al., in their study, did not observe a statistically significant association between the survival rate and microvascular density, analyzed by CD105, regardless of their value, both in de novo and recurrent cases [44]. Similar results were described by Mihic et al. [45]. In contrast, Behrem et al. found a statistically significant correlation between the two biomarkers in a numerically similar sample to our study [46].

Bastos et al. in their study on a cohort of patients treated with bevacizumab or temozolomide regardless of the measured microvascular density (CD105) did not record longer survival in any of the analyzed groups. They also did not observe differences between survival and location, and the Ki-67 index did not impact the prognosis [43]. Regarding survival, some data have shown a correlation between increased microvessel density, evaluated by CD105 immunostaining (MVD-CD105), and a worse survival/prognosis of patients with glioblastomas characterized by increased MVD-CD105 [32,46], while others have not shown any significant association in the studied cohorts [44,45].

Regarding the vascular invasion of glioblastomas, studies have shown differences between vascular patterns and MVD through CD105 immunostaining. Some authors suggest that cells in the infiltrative zone had a molecular composition showing the presence of immature vascular structures, alongside a minimal number of endothelial cells and low expression of VEGF receptors, compared to the tumor central zone where vessels had mature endothelial cells, implying a higher value for VEGF expression. This observation confers greater aggressiveness in tumors by MVD-CD105 from the central zone compared to the tumor periphery [47]. In contrast, Bastos et al. found no association between MVD-CD105 analyzed in the central zone or in the peripheral zone compared to overall survival [43]. Maddison et al. analyzed microvascular densities in both primary glioblastomas and recurrences and showed a decrease in total microvascular density, including endothelial cells analyzed by CD34 immunostaining, in recurrent glioblastomas. Moreover, they found a statistically significant decrease in terms of MVD-CD34 between de novo cases and recurrent cases [48].

Several clinical studies have investigated the activity of anti-VEGF monoclonal antibodies in glioblastomas, presenting their limitations regarding survival. Resistance mechanisms to antiangiogenic therapy, including vessel co-option or hypoxic signaling, are associated with the tumor microenvironment by modulating glioma stem cells [49]. The interactions between tumor glial cells and the tumor microenvironment, especially glioma stem cells, enhance the formation of neovascular structures from the tumor stroma, contributing to an unfavorable prognosis in terms of survival. Angiogenesis in glioblastomas is predominantly attributed to VEGF upregulation, which stimulates proliferation and migration or transdifferentiation of glioma cells into endothelial cells. Hypoxia-induced VEGF expression, due to the pronounced proliferation of tumor glial cells, subsequently stimulates the formation of poorly formed neovessels. However, these newly formed structures are not sufficient for adequate vascularization of tumor glial cells, so they migrate from hypoxic regions by invading healthy peritumoral tissue, while they may undergo transcriptional modifications that further increase resistance to therapy [50]. Some studies have shown that anti-VEGF treatment with bevacizumab did not yield the desired results in the recurrence of glioblastomas, which is inevitable in most cases. Therefore, another approach is needed regarding halting tumor angiogenesis. TRC105, a new chemotherapeutic agent, induces antibody-dependent cellular cytotoxicity and apoptosis of human vascular endothelial cells and tumor cells positive for endoglin and inhibits angiogenesis in response to VEGF [51]. Moreover, TRC105, being an antibody targeting CD105, seems to enhance the effect of bevacizumab in vivo, and is being considered an option for treatment with or without the administration of Bevacizumab [52].

## 4. Materials and Methods

### 4.1. Clinical Data

In this retrospective study, fifty-four patients diagnosed with glioblastoma at the Pathology Department of the Emergency County Clinical Hospital Târgu Mureș, between February 2014 and December 2017, were included. The inclusion criteria were as follows: (1) histopathological confirmation of glioblastoma without prior diagnosis or oncologic treatment for any type of brain tumor; (2) no history of brain biopsy; and (3) availability of tumor tissue in at least two paraffin blocks for the determination of IDH1-R132H, ATRX, CD34, CD105, Ki67, and p53 immunoexpression. Histopathological diagnoses were re-evaluated by a neuropathologist in accordance with the World Health Organization’s classification of nervous system tumors elaborated in 2016.

### 4.2. Immunohistochemistry

Operative samples were fixed in formalin, embedded in paraffin, and subsequently sectioned at a thickness of 3 μm. The obtained sections followed the standard deparaffinization and rehydration procedure. Endogenous peroxidase was blocked by applying a 10 min treatment with 3% H_2_O_2_. Antigen retrieval was performed by steam heat treatment for 25 min in a citrate buffer solution (pH 6). We used mouse monoclonal antibody IDH1R132H, IHC132 clone (BioSB, Santa Barbara, CA, USA), diluted at 1:25, hPh, for 60 min; mouse monoclonal antibody ATRX, BSB-108 clone (BioSB, Santa Barbara, CA, USA), diluted at 1:50, hPh, for 60 min; rabbit monoclonal antibody CD34, EP88 clone (BioSB, Santa Barbara, CA, USA), diluted at 1:100, hPh, for 60 min; rabbit monoclonal antibody CD105, EP274 clone (BioSB, Santa Barbara, CA, USA), diluted at 1:200, hPh, for 60 min; mouse monoclonal antibody Ki67, MM1 clone (Novocastra, Leica Biosystems, Deer Park, IL, USA), diluted at 1:150, hPh, for 60 min; and mouse monoclonal antibody p53, DO7 clone (BioSB, Santa Barbara, CA, USA), diluted at 1:800, hPh, for 60 min. The EnVision Flex/peroxidase herring system (HRP) (Dako, Santa Clara, CA, USA, 30 min) was used for signal amplification, and 3,3’-diaminobenzidine (DAB) chromogen was used for primary antibody detection. Subsequently, slides were counterstained with hematoxylin.

### 4.3. Slide Evaluation

A preliminary examination of the slides was performed using an Olympus BX46 microscope, and then the slides were scanned with a 3DHistech PANORAMIC 1000 scanner. Microvascular density was determined based on CD34 and CD105 immunoexpression. In the tumor tissue, four areas with the highest densities of blood vessels were selected, initially with a low-power objective (×40) and then using a higher-power objective (×400). Objective quantification of the percentage of microvascular density in the tumor stroma and in the normal brain tissue located next to the tumor tissue was performed using the Slideview program (3DHistech, Budapest, Hungary), and the mean values obtained for the four analyzed areas were calculated. We considered immunohistochemical reaction positive if solitary or grouped endothelial cells, whether or not involved in vascular lumen formation, showed a positive reaction. The endogen immunohistochemical control was considered for CD34 and CD105 of the endothelial cells in normal brain tissue vessels [53,54].

*IDH1* mutation expression was determined by quantifying positively stained cytoplasmic tumor cells, regardless of color intensity. Cases in which ≥10% of cells were stained were defined as positive (*IDH1* mutant), while cases where this value did not exceed 10% of tumor cells were considered negative (*IDH1* wild type) [55].

For the *ATRX* marker, *ATRX* gene mutations are followed by a loss of nuclear immunoexpression in over 50% of tumor cells (*ATRX* loss–*ATRX* mutant type). If *ATRX* immunoexpression remains preserved in over 50% of tumor cells, the respective tumor is considered the *ATRX* wild type, with the endogenous positive control being endothelial cells [55].

The Ki-67 proliferation index was determined as the percentage of positively stained tumor cells (regardless of intensity) relative to 1000 cells. The presence of *p53* was determined using the percentage of immunopositive cells relative to 200 cells in 5 fields. We considered negative immunoexpression if immunostaining was present in <10% of tumor cells (wild type) and positive if >10% of examined tumor cells were immunopositive (mutant type) [55].

The interpretation of immunohistochemical results was supervised by a neuropathologist.

### 4.4. Statistical Analysis

Descriptive and inferential statistics were performed. The normality of the distribution of continuous variables was tested using the Shapiro–Wilk test. Continuous variables were expressed as a median (25th percentile, 75th percentile), and medians were compared using the Mann–Whitney test. Categorical variables were displayed as numbers, and between-group comparisons were performed using the chi-square test. A value of *p* < 0.05 was considered significant. The IBM SPSS Statistics 22.0 program (IBM Corporation, Armonk, NY, USA) software was used for statistical analyses.

### 4.5. Study Limitation

The main limitation of our study is represented by the small sample size.

### 4.6. Ethics Committee

This study was approved by the Ethics Committee of the County Emergency Clinical Hospital Târgu Mureș (24494/16.10.2020).

## 5. Conclusions

This study has highlighted a variety of percentage ranges of vascular microdensity evaluated by CD34 and CD105 immunohistochemical expressions, without their correlation with *p53* and Ki-67 in primary or secondary glioblastomas, in the studied geographical area. Since multidisciplinary therapeutic strategies, especially antiangiogenic ones, are under evaluation and standardization for glioblastomas, further targeted molecular studies are needed in the future.

## Figures and Tables

**Figure 1 ijms-25-06810-f001:**
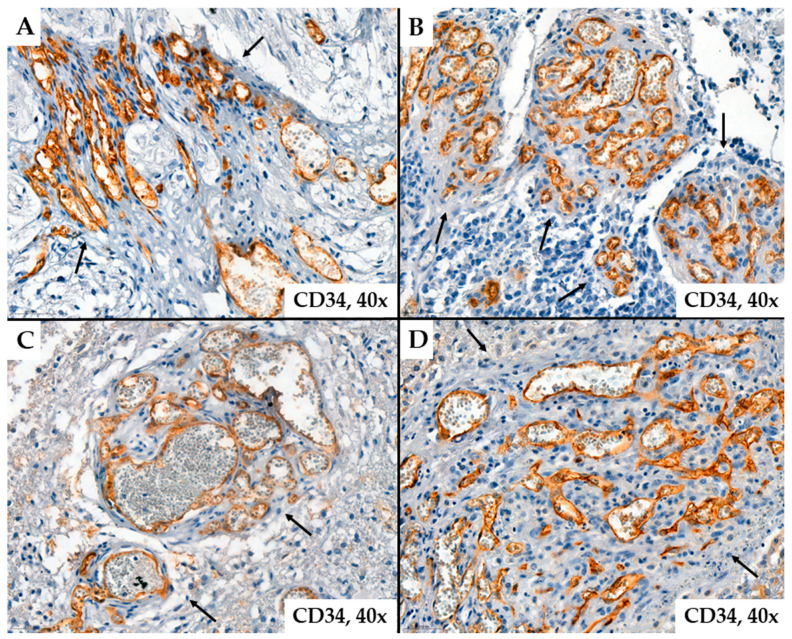
Representative cases of neovessels in glioblastomas with CD34 ((**A**,**B**)*—*patient #16, female, 59 years of age, (**C**,**D**)*—*patient #29, female, 58 years of age) immunohistochemistry stain. Several aspects can be observed regarding the morphology of the examined vessels: they are organized or grouped in glomeruloid structures (arrows), have variable shapes and sizes, exhibit unequal lumens, and show endothelial cells with cyto-nuclear atypia, typically oriented towards the vessel lumen.

**Figure 2 ijms-25-06810-f002:**
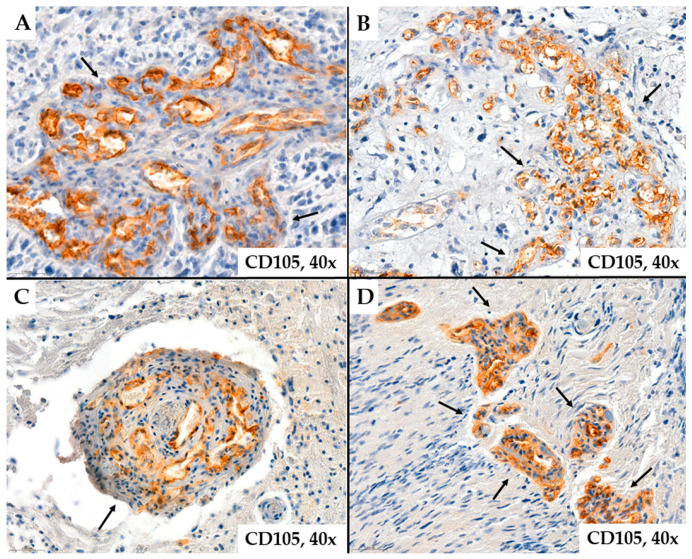
Representative cases of neovessels in glioblastomas with CD105 ((**A**)*—*patient #46, male, 46 years of age, (**B**)*—*patient #47, male, 67 years of age, (**C**)*—*patient #52, male, 70 years of age, (**D**)*—*patient #14, female, 63 years of age) immunohistochemistry stain. Several aspects can be observed regarding the morphology of the examined vessels: they are organized or grouped in glomeruloid structures (arrows), have variable shapes and sizes, exhibit unequal lumens, and show endothelial cells with cyto-nuclear atypia, typically oriented towards the vessel lumen.

**Figure 3 ijms-25-06810-f003:**
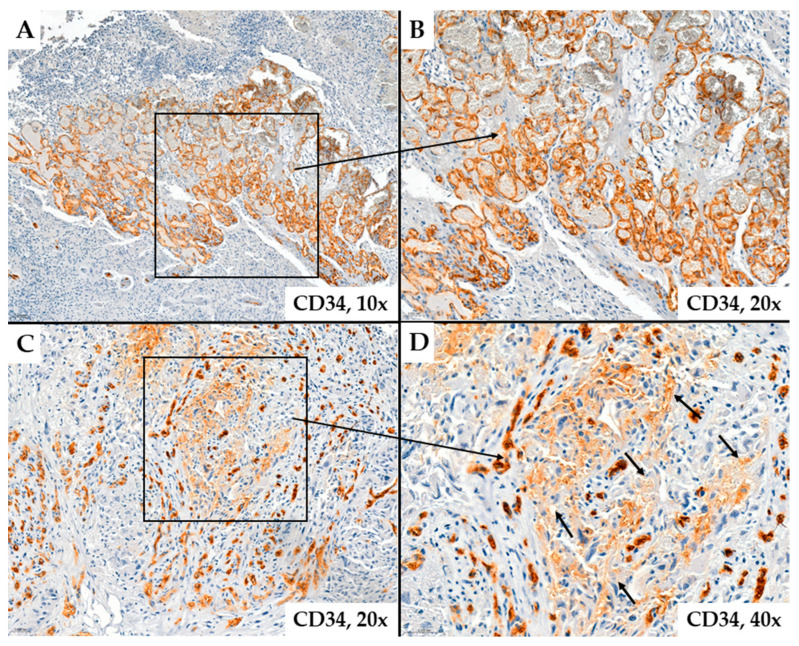
Abundant microvascular density in glioblastomas evidenced by CD34 immunohistochemical staining. Image (**A**) shows a rich vascular network of different-sized vessels. Image (**B**) is an enlarged version of Image (**A**) (patient #54, female, 27 years of age). In Image (**C**) (patient #16, female, 59 years of age), different shades of immunohistochemical staining can be observed in endothelial cells and around these vessels; in some glial cells, a brownish coloration is visible (Image (**D**)) (arrows), likely due to endothelial transdifferentiation.

**Figure 4 ijms-25-06810-f004:**
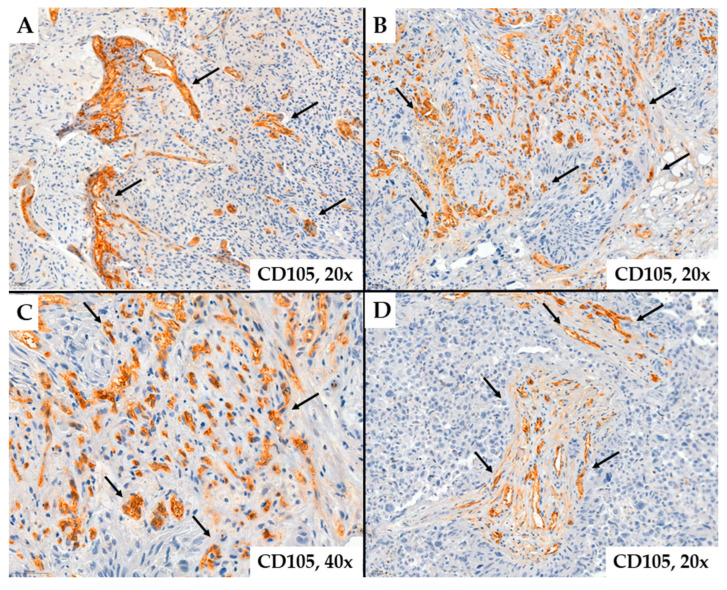
Microvascular density demonstrated by CD105 ((**A**)*—*patient #8, male, 45 years of age, (**B**–**D**)*—*patient #54, female, 27 years of age) in glioblastomas. The images suggest multiple variations in the shapes and sizes of the vascular lumens suggesting that these vascular structures are neoformed (arrows).

**Figure 5 ijms-25-06810-f005:**
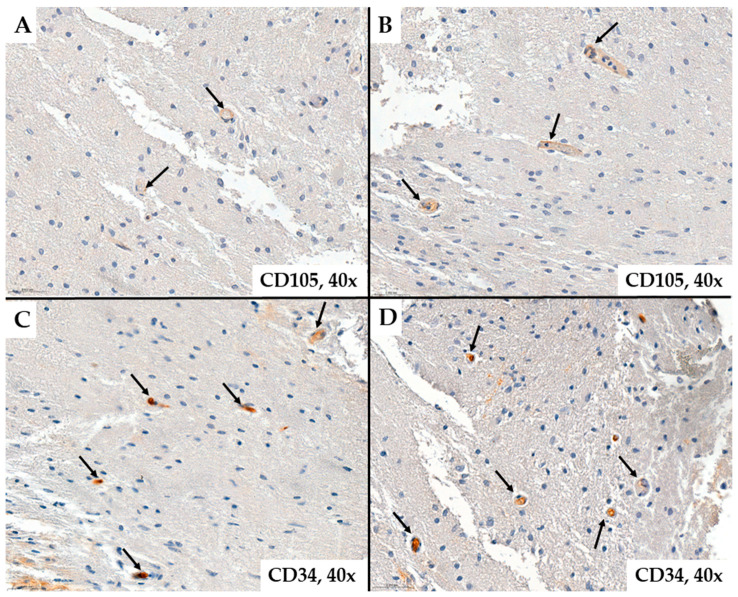
Normal brain tissue colored by CD105 ((**A**,**B**)*—*patient #54, female, 27 years of age) and CD34 ((**C**,**D**)*—*patient #31, male, 74 years of age) immunostaining. It can be seen that the number of vessels (arrows) is significantly lower than the tumor tissue in Figure 1, Figure 2, Figure 3 and Figure 4.

**Figure 6 ijms-25-06810-f006:**
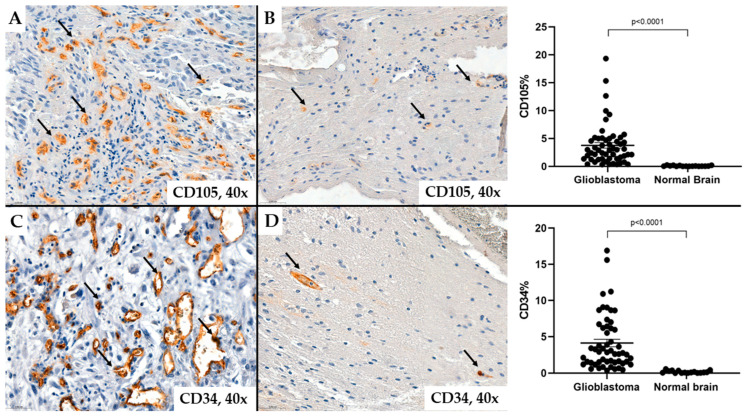
The percentage of microvascular density in glioblastomas and normal brain tissue evaluated by CD34 ((**C**,**D**)*—*patient #46, male, 46 years of age) and CD105 ((**A**,**B**)*—*patient #54, female, 27 years of age). The images suggest the number, shape, and size of vessels (arrows) in normal brain tissue compared with the glioblastoma tissue.

**Figure 7 ijms-25-06810-f007:**
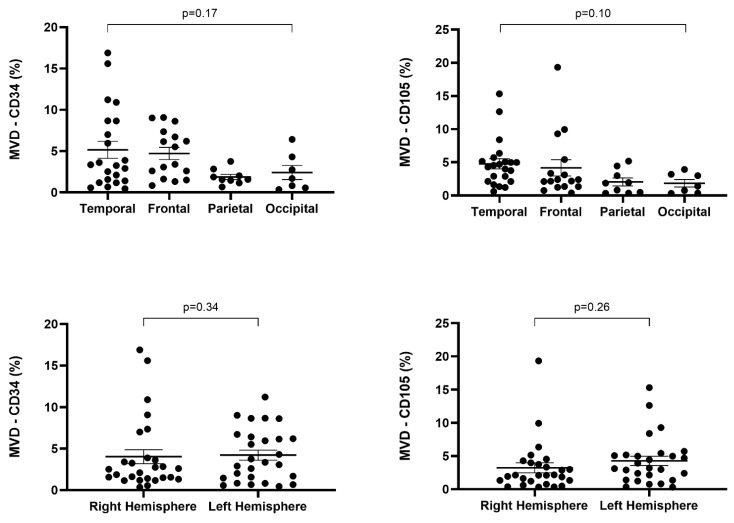
Correlation between microvascular density by CD34 or CD105 and laterality/orientation in glioblastomas.

**Figure 8 ijms-25-06810-f008:**
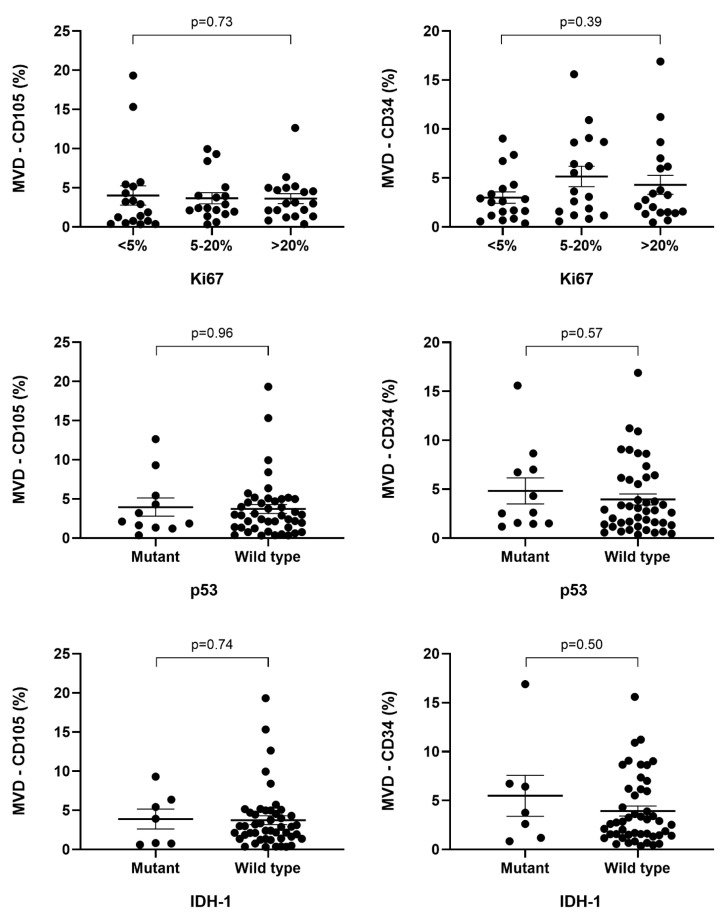
Correlation between microvascular density by CD34 or CD105 and IDH1, p53, and Ki67 in glioblastomas.

**Figure 9 ijms-25-06810-f009:**
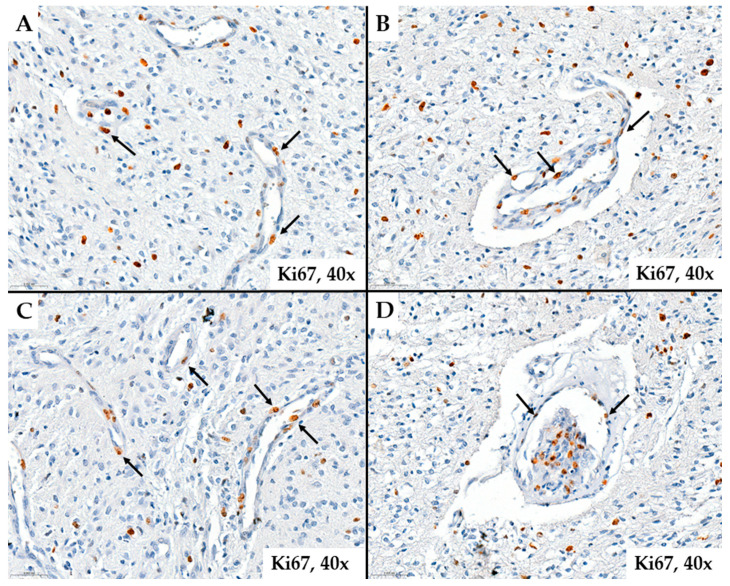
Ki67 index marks endothelial cells (arrows), suggesting endothelial cell proliferation probably derived from transdifferentiated glial cells ((**A**–**D**))*—*patient #37, male, 41 years of age).

**Table 1 ijms-25-06810-t001:** MVD-CD34 vs. MVD-CD105 in relation to clinico-pathological parameters and *IDH1* mutation, *p53, ATRX*, respectively, with the Ki67 proliferation index in glioblastomas.

N (nr)	MVD-CD34				MVD-CD105			
	<2%	2–5%	>5%	*p*	<2%	2–5%	>5%	*p*
**AGE**								
<50 years	7	5	7	0.24	7	6	6	0.54
50–65 years	4	8	5		6	7	4	
>65 years	10	3	5		6	10	2	
**GENDER**								
Male	11	9	9	0.97	8	15	6	0.31
Female	10	7	8		11	8	6	
**LOCALIZATION** (lobe)								
Frontal	4	4	8	0.17	5	7	4	0.10
Temporal	7	7	8		4	11	7	
Parietal	6	3	0		6	3	0	
Occipital	4	2	1		4	3	0	
**LATERALITY**								
Right hemisphere	12	9	6	0.34	12	11	4	0.26
Left hemisphere	9	7	11		7	12	8	
***IDH1* MUTATION**								
*IDH1* Wild Type	19	14	14	0.75	16	22	9	0.2
*IDH1* Mutant Type	2	2	3		3	1	3	
**p53 MUTATION**								
Mutant type	4	3	4	0.92	5	3	3	0.51
Wild type	17	13	13		14	20	9	
**Ki67**								
<5%	8	7	3	0.38	9	4	5	0.26
5–20%	6	3	8		5	8	4	
>20%	7	6	6		5	11	3	
** *ATRX* **								
Wild type	17	11	11	0.5	14	19	6	0.12
Mutant type	4	5	6		5	4	6	

**Table 2 ijms-25-06810-t002:** Correlation between mean values of MVD-CD34 and MVD-CD105 with respect to IDH1, ATRX, p53, and Ki67 index in glioblastomas.

	IDH1			
	Mutant	Wild		*p*
CD34%	3.75 (1.89–6.57)	2.76 (1.48–6.05)		0.503
CD105%	3.92 (0.80–5.90)	2.89 (1.38–4.63)		0.748
	p53			
	Mutant	Wild		*p*
CD34%	2.6 (1.53–6.86)	2.84 (1.36–6.05)		0.577
CD105%	2.12 (1.48–4.87)	2.9 (1.36–4.84)		0.966
	ATRX			
	Mutant	Wild		*p*
CD34%	3.75 (2.11–7.69)	2.53 (1.43–5.74)		0.105
CD105%	3.92 (1.03–6.04)	2.4 (1.38–4.50)		0.39
	Ki67			
	<5%	5–20%	>20%	*p*
CD34%	2.56 (1.16–3.89)	3.62 (1.57–8.63)	2.76 (1.48–6.05)	0.393
CD105%	2.39 (0.76–5.15)	2.39 (1.95–3.99)	2.99 (1.73–4.84)	0.73

## Data Availability

All data produced here are available upon request.

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
