# Peer review of "Evaluation of Microvascular Density in Glioblastomas in Relation to p53 and Ki67 Immunoexpression"

_ijms, 2024, doi:10.3390/ijms25126810_

Round 1

Reviewer 1 Report

Comments and Suggestions for Authors

The paper titled: “Evaluation of Microvascular Density in Glio-Blastomas in Relation to P53 and Ki67 Immuno-Expression” describes the histological quantification of angiogenesis through tumor microcirculation and potential correlation with ki67 and p53 protein expression.

The question is interesting and could potentially attract readers' attention.

The question posed is not novel, but it is interesting and could potentially attract attention for the readers. The results are well presented, even if they do not reach significance.

Also, the sections are well organized and lead the reader to the discussion of the scientific publication

In summary, this manuscript presents an interesting topic in the field of cancer biology with particular attention for the microcirculation in brain cancer and it is suitable for publication following minor revision.

Minor comments

- Why do the authors use the word glio-blastomas in the title?

- References 1 to 10 seem wrong, it would be better to replace them with references that report epidemiological data.

Author Response

Dear Reviewer,

Thank you for your time and patience. We appreciate your advice on correcting this manuscript, particularly your suggestions on editing the title, incorporating more updated references, and referring to epidemiological data. We have significantly revised the first paragraph of the introduction to reflect these changes. Below is the revised version, ensuring academic rigor and clarity.

We hope these changes meet your expectations and enhance the quality of our manuscript. Thank you once again for your valuable feedback.

Sincerely,

Dr. Sipos Tamas - Csaba

Reviewer 2 Report

Comments and Suggestions for Authors

This manuscript entitled " Evaluation of Microvascular Density in Glioblastomas in Relation to P53 and Ki67 Immuno-Expression" provides interesting information regarding variety of percentage intervals of microvascular density in primary and secondary glioblastomas using immunohistochemical markers CD34 and CD105, respectively. However, authors need to work more to improve the quality of this manuscript. Following are my concerns:

1.) In this study authors included fifty-four patients diagnosed with glioblastoma. My concern is how authors compare their finding with control group for immunohistochemical markers CD34 and CD105. Therefore, control healthy group is missing in this study.

2.) Please change images and include good quality pictures.

3.) Please include any statistical difference observed in bar graphs as well, which is not showing in bar graphs.

Comments on the Quality of English Language

Moderate editing of English language required

Author Response

Dear reviewer,

Thank you for your time and patience. We appreciate your advice on proofreading this manuscript, especially your suggestions on editing the graphics and images respectively. We have edited the requested control group by percentage analysis of normal brain tissue vessels. We have revised this in the material and method and in the results to reflect these changes. Below is the revised version, ensuring academic rigour and clarity.

We hope these changes meet your expectations and improve the quality of our manuscript. Thank you once again for your valuable feedback.

Sincerely,

Dr. Sipos Tamas - Csaba

Round 2

Reviewer 2 Report

Comments and Suggestions for Authors

Thank you authors for providing this revised version of manuscript entitled "Evaluation of Microvascular Density in Glio-Blastomas in Relation to p53 and Ki67 Immuno-Expression." Following are my concerns:

1.) Please improve the format for abstract. Please write abstract in a paragraph without headings for methods, results and conclusion.

2.) Please mark the all IHC images with arrow as shown in figure 3 or 9. 

3.) Please include the graphs for staining as shown in this following paper.

Tumor Microenvironment and Microvascular Density in Human Glioblastoma.

 file:///Users/geetika/Downloads/Tumor_Microenvironment_and_Microvascular_Density_i.pdf

And please justify how your study is different from the study explained in this paper.

Comments on the Quality of English Language

 Moderate editing of English language required.

Author Response

Dear Reviewer,

Thank you very much for the suggestions provided to improve our manuscript. We have made the requested changes according to your specifications.

Best regards
Sipos Tamas-Csaba

Round 3

Reviewer 2 Report

Comments and Suggestions for Authors

Thank you authors for working on the comments to improve the quality of manuscript.

Comments on the Quality of English Language

Minor grammatical error need to check.

Author Response

Dear Reviewer,

I hope this message finds you well.

I am writing to express my sincere gratitude for the time and effort you have dedicated to revising our manuscript. Your insightful feedback and meticulous attention to detail have significantly enhanced the quality of our work.

We greatly appreciate your expertise and the thoughtful suggestions you provided. Your contributions are invaluable, and we are confident that these revisions will strengthen the manuscript substantially.

Thank you once again for your commitment and support. We look forward to any further guidance you may offer and remain open to additional feedback.

Warm regards,

Dr. Sipos Tamas Csaba
